# Poly(ADP-Ribose) Polymerase-1 Lacking Enzymatic Activity Is Not Compatible with Mouse Development

**DOI:** 10.3390/cells12162078

**Published:** 2023-08-16

**Authors:** Tatiana Kamaletdinova, Wen Zong, Pavel Urbánek, Sijia Wang, Mara Sannai, Paulius Grigaravičius, Wenli Sun, Zahra Fanaei-Kahrani, Aswin Mangerich, Michael O. Hottiger, Tangliang Li, Zhao-Qi Wang

**Affiliations:** 1Leibniz Institute on Aging—Fritz Lipmann Institute (FLI), 07745 Jena, Germany; tatiana.kamaletdinova@leibniz-fli.de (T.K.); pavel.urbanek@leibniz-fli.de (P.U.); mara.sannai@leibniz-fli.de (M.S.); paulius.grigaravicius@leibniz-fli.de (P.G.); zahra.fanaeikahrani@leibniz-fli.de (Z.F.-K.); 2State Key Laboratory of Microbial Technology, Shandong University, 72 Binhai Road, Qingdao 266237, China; wenzong@sdu.edu.cn (W.Z.); 202212538@mail.sdu.edu.cn (S.W.); swl19972023@163.com (W.S.); li.tangliang@sdu.edu.cn (T.L.); 3Molecular Toxicology, Department of Biology, University of Konstanz, 78464 Konstanz, Germany; mangerich@uni-potsdam.de; 4Nutritional Toxicology, Institute of Nutritional Science, University of Potsdam, 14469 Potsdam, Germany; 5Department of Molecular Mechanisms of Disease, University of Zürich, 8057 Zürich, Switzerland; michael.hottiger@dmmd.uzh.ch; 6Faculty of Biological Sciences, Friedrich Schiller University of Jena, 07743 Jena, Germany

**Keywords:** PARP1, PARP2, catalytic activity, development, ES cell differentiation, genotoxic stress

## Abstract

Poly(ADP-ribose) polymerase-1 (PARP1) binds DNA lesions to catalyse poly(ADP-ribosyl)ation (PARylation) using NAD+ as a substrate. PARP1 plays multiple roles in cellular activities, including DNA repair, transcription, cell death, and chromatin remodelling. However, whether these functions are governed by the enzymatic activity or scaffolding function of PARP1 remains elusive. In this study, we inactivated in mice the enzymatic activity of PARP1 by truncating its C-terminus that is essential for ART catalysis (PARP1^ΔC/ΔC^, designated as PARP1-ΔC). The mutation caused embryonic lethality between embryonic day E8.5 and E13.5, in stark contrast to PARP1 complete knockout (PARP1^−/−^) mice, which are viable. Embryonic stem (ES) cell lines can be derived from PARP1^ΔC/ΔC^ blastocysts, and these mutant ES cells can differentiate into all three germ layers, yet, with a high degree of cystic structures, indicating defects in epithelial cells. Intriguingly, PARP1-ΔC protein is expressed at very low levels compared to its full-length counterpart, suggesting a selective advantage for cell survival. Noticeably, PARP2 is particularly elevated and permanently present at the chromatin in PARP1-ΔC cells, indicating an engagement of PARP2 by non-enzymatic PARP1 protein at the chromatin. Surprisingly, the introduction of PARP1-ΔC mutation in adult mice did not impair their viability; yet, these mutant mice are hypersensitive to alkylating agents, similar to PARP1^−/−^ mutant mice. Our study demonstrates that the catalytically inactive mutant of PARP1 causes the developmental block, plausibly involving PARP2 trapping.

## 1. Introduction

Poly(ADP-ribose) polymerase 1 (PARP1) is the founding member of the ADP-ribosyltransferase (ART) superfamily [1,2,3]. PARP1 catalyses poly(ADP-ribosyl)ation (PARylation) on many acceptor proteins, mainly itself, which is known as auto-PARylation. Its C-terminus contains a catalytic domain named the ART domain with a specific β-α-loop-β-α architecture and forms a catalytic triad motif, which binds NAD^+^ for catalysis [4]. Catalytic glutamate of the HYE motif of the PARP signature ensures polymer chain elongation and branching [5]. PARP1 and PARylation have been shown to play multiple roles in DNA damage response (DDR), replication stress, transcriptional regulation, chromatin remodelling, cell death, RNA biogenesis, ribosome activity, and lipid metabolism. These roles have been attributed to its enzymatic and/or scaffold functions, although it is often unclear [6,7,8,9,10].

After binding to damaged DNA, PARP1 catalyses PARylation to form poly(ADP-ribose) (PAR), which signals and recruits other repair factors to perform DNA repair [9,11,12]. The PAR chain synthesis is swift [13,14], and the degradation of PAR chains is also very rapid [15]. The lack of PAR-degrading enzyme activity that influences PAR homeostasis is not compatible with life: for example, complete knockout (KO) of PAR glycohydrolase (PARG) causes embryonic lethality at day 3.5 (E3.5) [16], although mice lacking only the nuclear isoform (PARG_110_) are viable [17]. Thus, coordinated PAR chain synthesis and degradation serve proper signalling and response [18,19]. Non-covalent PAR binding by diverse PAR readers is also actively involved in cellular metabolism and initiates a signalling network within the cell [11,20,21,22]. Thus, PAR readers serve as another layer of cellular signalling.

It has been well established that PARP1, as a scaffold protein, is required for p65-mediated activation of NF-κB [8], leading to the transcriptional activation of genes encoding inducible nitric oxide synthase (iNOS), P-selectin, cytokines, and growth factors [8,23]. This process does not seem to require the enzymatic activity of PARP1. In a complex with the nuclear respiratory factor 1 (NRF1), PARP1 serves as a transcriptional co-activator, without engaging its enzymatic activity. However, when PARP1 is catalytically active, it causes transcriptional repression of NRF1-dependent promoters [7]. Apart from its co-factor role in transcriptional machinery, PARP1 may directly regulate transcriptional elongation controlling RNA polymerase II (RNAPII) pausing via nucleosome deposition and post-translational modifications of histones at exon–intron connections [6]. It has been shown that PARP1 binds nucleosomes at their linker region and promotes their reorganisation [24].

PARP1 complete knockout (KO, PARP1^−/−^) mice are viable and do not show any visible abnormalities at birth and during their lifespan [25,26,27]. However, these mutant mice show increased sensitivity to acute genotoxic stress [13,26,28,29]. PARP1^−/−^ cells show full characteristics of genomic instability, such as an increased frequency of chromosome breaks, fusions, telomere shortening, and aneuploidy [30], as well as an increased sensitivity to alkylating agents [31]. These are believed to be a consequence of lacking PARP1 catalytic activity. On the other hand, PARP1^−/−^ mutant mice are resistant to septic shock induced by lipopolysaccharide (LPS) or bacterial caecal ligation and needle puncture (CLP) [32,33] because of a decreased release of inflammatory cytokines due to lacking PARP1 as a co-factor for NF-κB transactivation [34]. In addition, PARP1^−/−^ mutant mice are resistant to ischemic brain injury [35,36], streptozotocin-induced diabetes [25,37], ischemic renal injury [38], and traumatic brain injury [39].

An attempt to separate the enzymatic function from its protein scaffold has been made to modulate the levels of PARylation activity of PARP1. For instance, knocking in a point mutation of PARP1 (PARP1-D993A) has impaired its enzyme kinetics and the complexity of PAR chains, thereby indicating hypo-PARylation [13]. Homozygous PARP1-D993A mutant mice are viable and do not show any visible abnormalities during their lifespan. PARP1-D993A mutant cells showed only mild impairment of base excision repair (BER) and DNA damage response (DDR). Intriguingly, PARP1-D993A mutant animals are hypersensitive to alkylating agents and behave similarly to PARP1^−/−^ mice [13].

These harmful and beneficial effects of PARP1 null mutation are thought to be attributed to the loss of both enzymatic and scaffold functions of PARP1 [10]. The dualistic ability of PARP1 to function as an enzyme and a scaffold protein simultaneously or alternately has accumulated many contradictory explanations of PARP1/PARylation. Here, we aimed to dissect the catalytic-activity-dependent and independent functions of PARP1 in vivo by inactivating its catalytic activity while retaining the PARP1 protein. In contrast to the complete knockout of PARP1, catalytically dead PARP1 in mice causes embryonic lethality. If the inactivation of the catalytic activity is introduced in adulthood, surprisingly, mutant mice remain viable but exhibit hypersensitivity to genotoxic stress.

## 2. Materials and Methods

### 2.1. Generation of PARP1^+/ΔC^ Mice

The sgRNA was designed with the CRISPR design tool (http://crispr.mit.edu/, accessed on 11 January 2015) (AAG GAT ACT CAT TAT ACA GC AGG) and produced in vitro by U6-RNA transcription as previously described [40] using the primer GCA GCT AAT ACG ACT CAC TAT AGG GAA GGA TAC TCA TTA TAC AGC GTT TTA GAG CTA GAA ATA G and the plasmid px330 (Addgene, Watertown, MA, USA, #42230) as a template; RNA concentration was measured with Qubit 2.0. The Cas9 mRNA (Transfection-ready Cas9 SmartNuclease mRNA (Eukaryotic version) #CAS500A-1, System Bioscience, Palo Alto, CA, USA) and the sgRNA were injected into the cytoplasm of one-cell-stage mouse zygotes (C57BL/6JRj) in TE buffer (10 mM Tris-Cl pH8; 0.1 mM EDTA) at a working concentration of 50 and 25 ng/µL, respectively. A total of 24 h after injection, surviving zygotes were transferred into oviducts of B6CBAF1 pseudo-pregnant females. One pair of primers amplifying a 685 bp region around the sgRNA target site was used for the screening of the locus F0 and F1 generation by Sanger sequencing: Parp1_screening Fwd: GAA TAC CAG GAA CCA AGT CAG G; Parp1_screening_Rev: GAG TCC CCA AGA TCT CTT ACC C. All sequences are given from 5′ to 3′ orientation. Genotyping and sequencing revealed an 8 bp deletion in exon 22, causing the loss of the 33 amino acids at the very C-terminus of PARP1. The mutation transmitted through the mouse germline, and thereby the PARP1^/+ΔC^ mouse line was generated.

### 2.2. Mice

PARP1^+/ΔC^, PARP1^−/−^ (Parp1^tm1Zqw^) [27], Artd1^flox/flox^ (B6-ARTD1(loxP)^tm1PG^) [41], Rosa26-CreERT2 (Gt(ROSA)26Sortm1(cre/ERT2)Tyj), and PARP1^ΔC/flox^;Rosa26-creER mouse lines were bred and housed in the Leibniz Institute on Aging—Fritz Lipmann Institute (FLI, Jena, Germany). All animal work at the FLI was conducted according to German animal welfare legislation and approved by the Thüringer Landesamt für Verbraucherschutz (TLV) (license numbers: §11, 03-042/16, FLI-21-014_2022_ZRA, and O_ZQW_22-24). Mouse strains of PARP1-ΔC were also maintained at the animal facility of Shandong University, Qingdao, P. R. China. Animal care and experiments were performed in accordance with the ethics committee’s guidelines (license number: SYDWLL-2022-037).

### 2.3. Tamoxifen Administration and Genotoxic Treatment of Mice

Six to eight weeks old PARP1^ΔC/flox^;Rosa26-creER mice were treated with tamoxifen (#T5648, Sigma-Aldrich, St. Louis, MO, USA) at 75 mg/kg/day by intraperitoneal injection for 4 consecutive days. Seven days after the last dose, the mouse heart, liver, spleen, lung, kidney, brain, and intestine tissues were genotyped. On the eleventh day after the initial tamoxifen injection, the mice were intraperitoneally injected with a single dose of MNU (150 mg/kg of body weight) or solvent (sham). The mice were scored daily, and the survival rates were evaluated. On the fifth day following the MNU injection, a group of mice was also sacrificed to conduct intestinal histology.

### 2.4. Genomic DNA Isolation and Genotyping

Genomic DNA (gDNA) was isolated either from the tail cut of 3- to 10-day-old mouse pups or from the cell pellets, according to [42]. Purified gDNA was utilised for the genotyping using following primer sets:*PARP1-ΔC (Str1): 1-PARP1-dCt-Wt-Fw*: GAA TAC CAG GAA CCA AGT CAG G, *dCt-F3*: TCT GGT GTC AAC GAT GCT, *PARP1-E988K-R1*: CTC AGG ACT AGT CTA GGC AA;*PARP1-KnockOut (PARP1-KO) (Str2*): *KO-ZQWstr-Fw-1*: AGG TCT ACG GGA CAC TTT AGG, *OVL-RI*: CCT TCC CAG AAG CAG GAG AAG, *NEO RII*: GCT TCA GTG ACA ACG TCG AG;*PARP1-FLOX (Str3): PARP1-flox-Fw*: GCT TCT ACT ACC TCC CAA GAA AGA GCG, *PARP1-flox-Rv:* GGC TTT AGT GTG GCA ACT TAT CCC, *PARP1-del-Rv*: CCT CTG CTG CGT GAC TAA GGC;*PARG-KO* (*Str4*): *TK2800*: TCC TTT TAT GTA GCT GCC TG, *TK3033*: GGT TAA CGT GAG GTT TAA AT, *R327*: CAC AAG TTC CAC GGA GAC CC;*Cre* (*Str5*): *Cre 1*: CGG TCG ATG CAA CGA GTG ATG), *Cre 2*: CCA GAG ACG GAA ATC CAT CGC, *B2-1*: CAC CGC AGA ATG GGA AGC CGA A, *B2-2*: TCC ACA CAG ATG GAG CGT CCA G.

All sequences are given from 5′ to 3′ orientation.

### 2.5. Plasmid Cloning and Vector Construction

The following plasmids were used to assemble the constructs for PARP1-FL and PARP1-ΔC levels reconstitution in PARP1^−/−^ cells: PARP1-cDNA human (Dharmacon Inc., Lafayette, CO, USA) was used as a template for PARP1-FL or PARP1-ΔC fragments amplification constructs; XLone-GFP (from Xiaojun Lian [43], #96930, 6327 bp, Addgene) was used as a backbone to design XLone-GFP-PARP1-ΔC (9285 bp) and XLone-GFP-PARP1-FL (9378 bp); MSCV-IRES-GFP (from Tannishtha Reya, #20672, 6488 bp, Addgene) was used as a backbone to design PARP1-ΔC-IRES-GFP (9446 bp) and PARP1-FL-IRES-GFP (9539 bp) for independent translation initiation by internal ribosome entry site (IRES) element. The following primers were used for assembling: (1) XLone-GFP-PARP1-ΔC and XLone-GFP-PARP1-FL—EGFP-PARP1-dCt-Fw: AAT GTA CAA GGA ATT CGG ATC GGG ATC GGC GGA GGC CTC GGA GAG GCT TTA T, EGFP-PARP1-full-Rv: TAT AAC TAG TTT ACC ACA GGG ATG TCT, PARP1-dC-EGFP-Rv: TAT AAC TAG TTT ATA CAG CAT CGT TGA CAC CAG A. (2) PARP1-ΔC-IRES-GFP and PARP1-FL-IRES-GFP—PARP1-h-unique-IRES-Fw: TAT ACT CGA GAT GGC CCA GTC TTC GGA TAA G, PARP1-full-h-IRES-Rv: TAT ACT CGA GTT ACC GGG AGG TCT TAA AAT TGA ATT TC, PARP1-DelC-h-IRES-Rv: TAT ACT CGA GTT AAG TGT CAT TCA CAC CAG ATG. All sequences are given from 5′ to 3′ orientation. Amplification was performed with CloneAmp HiFi PCR Premix (#639298, TaKaRa, Kusatsu, Shiga, Japan).

### 2.6. Cell Transfection

Transfection of cells was performed on a 6-well plate (stable cell line generation), 12-well plate (Western blotting), or T75 format (retrovirus production). All transfection experiments were performed with Lipofectamine^TM^ 3000 kit (#L3000008, Thermo Fisher Scientific, Waltham, MA, USA) according to the manufacturer’s protocols.

### 2.7. RT-qPCR

RNA isolation, reverse transcription (RT) and semi-quantitative RT-PCR was carried out as described previously [44] at a LightCycler^®^ 480 (Roche, Basel, Switzerland) using the following primers: *gapdh* Fw: GTG TTC CTA CCC CCA ATG TGT, Rv: ATT GTC ATA CCA GGA AAT GAG CTT, *pou5f1* Fw: CTT GGG CTA GAG AAG GAT GTG, Rv: CTG AGT AGA GTG TGG TGA AGT G, *nes* Fw: GAT GTC CCT TAG TCT GGA AGT G Rv: GGT CAG GAA AGC CAA GAG AA, *ctnnb1* Fw: TGC AGC TGG AAT TCT CTC TAA C, Rv: CCA CAA CAG GCA GTC CAT AA, *cdh2* Fw: GAA GAA GGT GGA GGA GAA GAA G, Rv: CCC AGT CAT TCA GGT AGT CAT AG. All sequences are given from 5′ to 3′ orientation. The expression levels of every gene were normalised to the GAPDH levels of the same sample.

### 2.8. Isolation of ES Cells and Primary MEF Cells

PARP1^+/ΔC^ blastocysts at the E3.5 embryonic stage were isolated and used to establish embryonic stem (ES) cell lines as described previously [17]. Each individual blastocyst was transferred into a cell culture dish with feeder cells. The blastocysts were cultured at 37 °C in a 21% O_2_ and 5% CO_2_ incubator until the establishment of ES cell lines. Primary MEF (pMEF) cells were isolated from E13.5 PARP1^+/ΔC^ embryos according to [13] and immortalised using p19ARF shRNA as previously described [45]. To generate PARP1^ΔC/ind−^ MEF cells, PARP1^flox/ΔC^ MEFs were treated with 1 μM 4-OHT for 15 days in culture to induce PARP1 deletion from the floxed allele, which was verified by PCR or Western blotting.

### 2.9. Cell Culture

Primary MEF, immortalised MEF, HeLa, and U2OS cells were cultured in embryonic fibroblast (EF) medium [DMEM (#42430082, Thermo Fisher Scientific) supplemented with 10% FBS (#S1810-500, Biowest, Nuaillé, France), 2 mM L-glutamine (#25030081, Thermo Fisher Scientific), 1 mM sodium pyruvate (#11360070, Thermo Fisher Scientific), 2 μM β-mercaptoethanol (#31350, Thermo Fisher Scientific), 50 units/mL penicillin, and 50 mg/mL streptomycin (Pen/Strep) (#15140122, Thermo Fisher Scientific)] according to [13]. ES cells were cultured in embryonic stem cells (ES) medium [DMEM supplemented with 15% FBS, 2 mM L-glutamine, 1 mM sodium pyruvate, 2 μM β-mercaptoethanol, 50 units/mL + 50 mg/mL of pen/strep, 1× MEM non-essential amino acids, and ESGRO^®^ Recombinant Mouse-LIF-Protein (#ESG1107, Sigma-Aldrich)] according to [46]. ES cells were cultured either on feeder cells or on 0.1% gelatine (#1.04078.1000, Merck, Darmstadt, Germany). All cells were cultured in a 5% CO_2_ incubator.

### 2.10. Proliferation Assay

10^5^ cells were plated onto a 0.1% gelatine layer on a 6-well plate with a Nunc^TM^ surface (#140675, Thermo Fisher Scientific), with the ES medium containing either DMSO (#41648, Sigma-Aldrich) or 2 μM Olaparib (AZD2281, #S1060, Selleckchem, Houston, TX, USA) that was renewed every day. Every four days, ES cells were counted, and 10^5^ ES cells were re-plated onto individual wells of a 6-well plate in the ES medium with either DMSO or Olaparib. A total of 3 × 10^5^ pMEF cells per well were plated on a 6-well plate with the EF medium with or without 4-OHT. Every three days, primary MEF cells were counted and re-plated.

### 2.11. Embryoid Body (EB) Formation Assay

The assay was performed as previously described [46]. In brief, 2 × 10^6^ ES cells were plated on 3 cm bacterial-grade Petri dishes in ES medium without LIF. The next day, cells were pelleted and resuspended in the ES medium without LIF. Then, the cell suspension was transferred into a 2 × 10 cm Petri dish. After 14 or 24 days, the EBs were fixed overnight with 4% paraformaldehyde (PFA) and sectioned for immunofluorescent staining. On days 0, 4, 8, 14, 24, and 36, a portion of the cells was also collected for RT-qPCR analysis.

### 2.12. Immunofluorescence

The EBs were cryosectioned at Microm HM550 cryostat. The immunofluorescence was performed according to [47]. Briefly, antigen retrieval was performed at 95 °C for 45 min in antigen retrieval buffer (1.8 mM citric acid; 8.2 mM sodium citrate; 0.05% Tween20^TM^ (#P9415, Sigma-Aldrich)) and then cooled until RT for 20 min. The sections were blocked in a blocking solution (1% BSA (#A2153, Sigma-Aldrich), 0.4% TritonX100 (#T8787, Sigma-Aldrich), 2.5% donkey serum, 2.5% goat serum in PBS). The primary antibodies were diluted in blocking buffer, applied to the sections, and incubated at 4 °C overnight. Secondary antibodies were diluted in blocking buffer, applied to the sections, and incubated at RT for 1 h. DNA was stained with 100 μg/mL DAPI. Alternatively, the sections were stained with haematoxylin and eosin dyes (HE). Slides were mounted with ProLong antifade mounting medium (#P36930, Thermo Fisher Scientific). Imaging was performed on an ApoTome.2 Imager.Z2 (Zeiss, Jena, Germany) or on an AxioScan Z.1, Colibry7 camera (Zeiss, Germany) (HE staining).

Immortalised MEF cells were fixed with 4% PFA solution for 20 min at 4 °C followed by incubation with 0.1% Triton-X100 solution in PBS for 10 min and blocked with blocking buffer (1% BSA in PBS) for 30 min. The primary antibodies were diluted in a blocking buffer and then applied onto the coverslip for 4 °C overnight incubation. The secondary antibody was diluted in PBST, applied onto the coverslips, and incubated at RT for 1 h. The nuclei were stained with 100 μg/mL DAPI, and the slides were further mounted with the ProLong Gold antifade mounting medium. Imaging was performed on an ApoTome.2 Imager.Z2.

The following antibodies and dilutions were used for immunofluorescence staining: Nestin 1:200 (#MAB353, Millipore, Burlington, MA, USA), N-Cadherin 1:300 (#610920, BD Transduction laboratories, Franklin Lakes, NJ, USA), β-Catenin 1:300 (#sc-7963, E-5, Santa Cruz, Santa Cruz, CA, USA), PAR 1:200 (#4336-BPC-100, Trevigen, Gaithersburg, MD, USA), anti-mouse Cy3 1:200 (#C2181-1Ml, Sigma-Aldrich), anti-rabbit Cy3 1:200 (#C2306-1Ml, Sigma-Aldrich), anti-Mouse Cy5 1:200 (#A10524, Invitrogen, Waltham, MA, USA), anti-rabbit Cy5 1:200 (#711-175-152, Jackson Immuno Research, West Grove, PA, USA), Alexa Fluor 488 donkey anti-Mouse IgG (H+L) 1:200 (#A11001, Invitrogen), and anti-rabbit Cy5 1:200 (#711-175-152, Jackson Immuno Research).

### 2.13. Genotoxic Treatment of ES Cells or MEFs

10^5^ MEF or ES cells were seeded in 12-well plates (#150200, Thermo Fisher Scientific) precoated with 0.1% gelatine (only for ES cells). The next day, the drug treatments were applied to the cells: 1 mM H_2_O_2_ (H1009, Sigma-Aldrich) in PBS for 15 min; 100 µM N-Methyl-N′-nitro-N-nitrosoguanidin (MNNG) (129941, Sigma-Aldrich), in EF medium without FCS and pen/strep for 2 h; 10 µM Olaparib alone (in complete EF medium) or as a supplement to H_2_O_2_ (in PBS) or to MNNG (in EF medium without FCS and Pen/Strep) for 10 min to 2 h.

### 2.14. Colony Formation Assay

Immortalised MEF cells were plated at low densities (4 × 10^5^ cells/well). The next day, DMSO (sham), MNNG 8 µM, or that supplemented with Olaparib (2 µM) were applied for 1 h. The following day, cells were re-passaged and grown in the EF medium for 5 days. Cells were fixed and stained with 0.5% crystal violet in 25% methanol for 30 min, washed with water, and air-dried. Quantification was performed using ImageJ. The surviving fraction was calculated after normalisation to the plating efficiency of untreated samples.

### 2.15. Fractionation for DNA-Trapping Analysis

ES cells were grown in T25 flasks (#136196, Thermo Fisher Scientific) precoated with 0.1% gelatine, and 1–2 × 10^6^ cells were incubated with either DMSO (sham), Talazoparib (#TA9H11E41C53, Targetmol Chemicals Inc., Boston, MA, USA, 100 nM, 15 min), MNNG (100 µM, 10 min), or a combination of both Talazoparib and MNNG in a 5-mL suspension in ES medium at 37 °C. For doubly-treated samples, cells were first incubated for 5 min with Talazoparib, and then MNNG was added for an additional 10 min. Following the treatments, cells were collected and fractionated using the Subcellular Protein Fractionation Kit (#78840, Thermo Fisher Scientific) according to the manufacturer’s instructions.

### 2.16. Western Blotting

Cells were lysed with lysis buffer according to [48]. A standard SDS-PAGE was performed. Proteins were transferred to PVDF membranes (#1620177, Bio-Rad Laboratories, Berkeley, CA, USA). The primary antibodies were diluted in TBST or blocking solution. The incubation was carried out overnight at 4 °C. The HRP-conjugated secondary antibodies were diluted in a blocking solution, and incubation was performed for 1 h at RT. Pierce^®^ ECL Western Blotting Substrate (#32106, Thermo Fisher Scientific) was used and the membrane was developed with Amersham Imager 600.

The following antibodies and dilutions were used for immunoblotting: PAR 1:1000 (#4336-BPC-100, Trevigen), GAPDH 1:5000 (#G8795, Sigma-Aldrich), PARP1 C-terminus 1:1000 (C2-10, #4338-MC-50, Trevigen), PARP1 N-terminus 1:1000 (#9542, Cell Signalling, Danvers, MA, USA), PARP2 1:1200 (#55149-1-AP, Proteintech, Rosemont, IL, USA), γ-tubulin 1:5000 (#T6557, Sigma-Aldrich), p53 1:1000 (#2524, Cell Signalling), LC3B 1:2000 (#L7543, Sigma-Aldrich), GFP 1:2000 (#2956, Cell Signalling), KPL HRP-goat-anti-mouse 1:5000 (#5220-0341, SeraCare, Milford, MA, USA), and KPL HRP-goat-anti-rabbit 1:5000 (#5220-0336, SeraCare).

### 2.17. Determination of NAD^+^ Levels

The ES cells were seeded on 96-well plates precoated with 0.1% gelatine. Half of the plate was used to determine the NAD^+^ levels with the NAD/NADH Cell-Based Kit (#600480, Cayman, Ann Arbor, MI, USA). A quarter of the plate was used for the cell counting and another quarter for the MTT assay. Briefly, 20 μL freshly prepared MTT solution (5 mg/mL stock solution in ddH_2_O) was added to the cells in 200 μL of complete medium for 2 h. The cell-bound MTT crystals were dissolved in 150 μL isopropanol for 30 min. The absorbance was measured at 570 nm.

### 2.18. Statistical Analysis

GraphPad Prism 8 was used for the statistical analysis. All the data were examined for normal distribution using quantile–quantile (QQ) plots. For the data with a normal distribution (Shapiro–Wilk test, *p*-value > 0.05), Student’s *t*-test was used to determine the statistical significance. If the data did not have a normal distribution (Shapiro–Wilk test, *p*-value < 0.05), a Mann–Whitney U-test was used to determine the statistical value. A one-way ANOVA non-parametric test was used for the comparison of more than one group. Finally, *p* < 0.05 was accepted as the significance level for all of the tests.

## 3. Results

### 3.1. Generation of PARP1^ΔC/ΔC^ Mice

To inactivate the PARylation capacity of PARP1, gene targeting in mouse germline was performed via the CRISPR/Cas9 gene editing technology, aiming to mutate the E988 residue (Appendix A), which has been shown to inactivate the enzymatic activity in biochemical assays [11,19,49], by changing GAG encoding E988 to AAG codon (K988) in exon 22 of the *Parp1* gene. One mosaic mouse from 15 newborns transmitted the mutation through the germline. Genotyping and sequencing revealed an 8 bp deletion in exon 22 located 11 bp upstream from the GAG codon encoding E988. This deletion led to the frameshift and introduction of a premature TAA stop codon in exon 22 (Figure 1A), causing the loss of the 33 amino acids at the very C-terminus of PARP1, designated herein PARP1-ΔC (Appendix A).

PARP1^ΔC/+^ mice appeared phenotypically normal and fertile (Appendix A). However, no newborn mice with the PARP1^ΔC/ΔC^ genotype were obtained among 232 offspring of heterozygote intercrosses (Figure 1B). We also did not find PARP1^ΔC/ΔC^ embryos among 40 genotyped embryos at embryonic day E13.5 (Figure 1B). Yet, we identified 7 embryos with the PARP1^ΔC/ΔC^ genotype (20.5%, according to the Mendelian ratio) after genotyping 34 embryos at E8.5 (Figure 1B), indicating that PARP1-ΔC embryos died between E8.5 and E13.5.

We obtained a cellular model by deriving embryonic stem (ES) cell lines from E3.5 embryos (blastocysts) from PARP1^+/ΔC^ intercrosses (Figure 1B). Out of the 74 ES cell lines established, 7 had PARP1^ΔC/ΔC^ genotype (9.5%), which was much lower than expected based on Mendelian ratios. This finding indicates a possible loss of the PARP1^ΔC/ΔC^ genotype during ES cell establishment. Moreover, it took almost twice as long to establish PARP1^ΔC/ΔC^ ES cells (PARP1-ΔC ES cells) compared to PARP1^+/ΔC^ or PARP1^+/+^ ES cells (Figure 1C), indicating that the PARP1-ΔC mutation compromises ES cell establishment. Nevertheless, once established, PARP1-ΔC ES cells proliferated at a rate comparable to other genotypes (Figure 1D).

### 3.2. Differentiation of PARP1^ΔC/ΔC^ ES Cells

To investigate how the PARP1-ΔC mutation blocked embryonic development and caused lethality, an embryoid body (EB) formation assay was performed. Immunofluorescence (IF) analysis of PARP1-ΔC EBs at 14 days revealed expression of lineage markers of all three germ layers, including Nestin for ectoderm, N-cadherin for mesoderm [46,50], and β-catenin for endoderm [51,52], all of which were similar to PARP1^+/+^ cells (Figure 2A). Furthermore, RT-qPCR analysis was performed to examine mRNA levels of these markers among EBs derived from PARP1^ΔC/ΔC^, PARP1^ΔC/+^, PARP1^+/+^, and PARP1^−/−^ ES cells at various stages of EB formation. While a significantly higher N-cadherin mRNA level was found in PARP1^−/−^ EBs compared to the other genotypes, PARP1-ΔC EBs expressed all these markers at similar levels comparable to control groups (Appendix A). Based on these findings, it can be inferred that PARP1-ΔC ES cells are able to differentiate into primary germ layers. However, at 24 days of EB formation, a significant increase in cyst-like structures was observed in PARP1^ΔC/ΔC^ EBs compared to PARP1^+/+^ (Figure 2B,C), suggesting a differentiation defect of epithelial-like cells, which may compromise the morphogenesis of organs [53].

### 3.3. Low Expression of PARP1-ΔC Protein

We next confirmed the C-terminal deletion of the PARP1-ΔC protein by performing Western blotting with an antibody against the C-terminal epitope of PARP1, which did not detect any signal from the PARP1-ΔC ES cells (Figure 3A, upper panel). The anti-PARP1 antibody that recognised epitopes in the N-terminal part of PARP1 (DEVD sequence) detected the presence of a truncated PARP1 protein in PARP1-ΔC ES cells, but at very low levels (Figure 3A, lower panel), as is evident in the overlay of two films of short and long exposures (Figure 3B). RT-qPCR analysis of PARP1-ΔC ES cells did not reveal significant differences in PARP1 mRNA levels of PARP1^+/+^, PARP1^ΔC/+^, and PARP1^ΔC/ΔC^ cells (Figure 3C). This indicates that the C-terminal truncation of PARP1 does not affect the transcription of the PARP1-ΔC allele or mRNA stability. The stability of the protein in ES cells was then examined using proteasome and lysosome inhibitors, MG132 and Bafilomycin A1 (BafA1), respectively. The level of wild-type PARP1 (+/+) and PARP1-ΔC proteins were found to increase by MG132, as controlled by p53, which is known to be degraded via the proteasomal pathway (Figure 3D,E). However, the fold increase in PARP1-ΔC protein after MG132 treatment was lower than that of PARP1-FL, judged by p53 levels in both cells. These findings suggest that PARP1-ΔC protein undergoes a proteasomal degradation pathway, but the low PARP1-ΔC protein level is likely mainly influenced by protein synthesis.

The presence of low levels of PARP1-ΔC protein implies a potential survival advantage for cells. To test this hypothesis, we tried to ectopically express GFP-tagged human PARP1-FL or PARP1-ΔC constructs in PARP1^−/−^ U2OS cells. Microscopy revealed that the GFP-PARP1-ΔC signals were significantly lower than that of the GFP-PARP1-FL or GFP-EV (empty vector) (Figure 3F). Western blot analysis further confirmed the low PARP1-ΔC expression in both PARP1^+/+^ and PARP1^−/−^ U2OS cells, indicating that the PARP1 (+/+) protein does not influence PARP1-ΔC protein expression (Figure 3G). To rule out the possibility that the low expression levels were inherent to the fusion with the GFP protein, we generated the constructs allowing independent translation of the GFP and PARP1 proteins by incorporating the internal ribosome entry site (IRES) element (Figure 3H). We transfected PARP1-FL-IRES-GFP, PARP1-ΔC-IRES-GFP, or IRES-GFP constructs into PARP1^+/+^ and PARP1^−/−^ HeLa cell lines and found that GFP levels, indicative of transfection efficiency, were comparable in both PARP1-FL and PARP1-ΔC transfectants (Figure 3H). However, PARP1-ΔC protein was only about 15% of the levels of ectopically expressed PARP1-FL as viewed by their expression in the PARP1^−/−^ background (Figure 3I), mimicking the situation of PARP1-ΔC ES cells, whereas PARP1-FL expression levels were comparable to endogenous PARP1. These experiments demonstrate that the PARP1-ΔC construct cannot be efficiently expressed, suggesting that the mutant protein is likely toxic to cells or downregulates its translation. The endogenous PARP1 in U2OS and HeLa cells did not affect the expression levels of—e.g., by trans-modifying—the enzymatic inactive mutant PARP1.

### 3.4. The PARylation Activity of PARP1-ΔC Cells

Due to the early embryonic lethality of PARP1^ΔC/ΔC^ mice, functional studies of PARP1-ΔC in vivo were impossible. To overcome this obstacle, inducible PARP1^ΔC/−^ (PARP1^ΔC/ind−^) mouse and cellular models were generated by breeding sequentially mice carrying the PARP1-ΔC allele with PARP1-flox mice (Artd1^fl/fl^, [41]) and CreER transgenic mice. Primary mouse embryonic fibroblasts (pMEFs) were isolated from E13.5 embryos, and cell lines with the PARP1^ΔC/fl^; CreER^tg/+^ genotype were obtained. When these cells were treated with 4-hydroxytamoxifen (4-OHT), the deletion of the PARP1-floxed allele was induced as confirmed by PCR genotyping to produce PARP1^ΔC/ind^^−^ allele (designated as inducible PARP1-ΔC or PARP1-iΔC) MEF cells (Appendix A). Of note, the protein levels of PARP1-ΔC in PARP1-iΔC MEF cells were barely detectable by Western blotting (Appendix A), similar to PARP1-ΔC ES cells.

Previous studies showed that Y986 and E988 are essential residues for the catalytic activity of PARP1 [54,55], and both residues are absent in PARP1-ΔC. We next examined the catalytic activity in PARP1-iΔC cells. First, we carried out IF staining to analyse PAR formation. Immortalised MEF cells were treated with either solvent (sham) or oxidative agent H_2_O_2,_ with or without PARP inhibitor (PARPi) Olaparib. Surprisingly, we detected a slight but significant increase in a nuclear PAR signal in PARP1-iΔC as well as in PARP1^ind−/ind−^ (inducible KO, or iKO) MEF cells after H_2_O_2_ treatment, but lower than wild-type controls (Figure 4A,B), indicating a residual PARylation activity in PARP1-iΔC cells, likely by other ARTs (see below). The PAR signal was much lower in PARP1-iKO MEF cells (Figure 4A,B). However, the PAR signal was not detectable by Western blotting in PARP1-iΔC MEF cells, nor in PARP1-iKO controls (Figure 4C). Intriguingly, the PARP2 levels were highly elevated in PARP1-iΔC MEF cells compared to that in PARP1^+/ind−^ and PARP1-iKO cells (Figure 4D,E). We hypothesised that, in PARP1-iΔC MEF cells, PARP2 forms dimers with PARP1 to facilitate PARylation, unlike the situation in PARP1^−/−^ cells, and that PARP1-iΔC itself lacks enzymatic activity.

PARylation results in the consumption of NAD^+^. We next measured NAD^+^ levels in mutant ES cells. Under steady-state conditions, NAD^+^ levels were approximately two-fold higher in PARP1-ΔC cells than in PARP1^+/+^ cells but were similar to those in PARP1^−/−^ ES cells (Figure 4F), indicating a limited capacity for PARylation of PARP1-ΔC ES cells. Treatment with H_2_O_2_ caused a reduction in NAD^+^ levels in all cell lines, but notably more in PARP1^+/+^ ES cells (Figure 4F). This further indicates that PARP1-ΔC cells have lower PARylation activity.

### 3.5. Trapping of PARP1 and PARP2 Proteins at Chromatin in PARP1-ΔC Cells

To investigate whether the trapping of PARP1-ΔC on chromatin could explain the cytotoxicity [9,56], PARP1^+/+^ and PARP1^ΔC/ΔC^ ES cells were treated with PARPi Talazoparib with or without alkylating agent MNNG. The cells were fractionated, and the chromatin-bound fractions were analysed by Western blotting. Similar to the PARP1-FL control, we did not detect prominent PARP1 protein in the chromatin fraction of PARP1-ΔC mutants under unstressed conditions, nor after Talazoparib treatment (Figure 5A,B). However, co-treatment of MNNG + Talazoparib enriched wild-type PARP1 protein in the chromatin-bound fraction, but failed to induce trapping of the PARP1-ΔC protein (Figure 5A,B). We also examined PARP2 levels on chromatin since they appear to be increased in PARP1-iΔC MEF cells. Strikingly, we found that in PARP1-ΔC ES cells, PARP2 was constantly present on chromatin in constitutively high amounts even under the steady-state status, in contrast to PARP1^+/+^ cells (Figure 5A,C). Of note, genotoxic treatment together with PARPi did not increase further the association of PARP2 with chromatin (Figure 5C), strongly suggesting that the PARP2 binding to chromatin, even at unstressed conditions, is saturated in PARP1-ΔC ES cells.

To further investigate the effect of PARP1 activity and its related trapping on viability, ES cells were cultured for eight passages in the presence of PARPi Olaparib. Olaparib completely inhibited the proliferation of PARP1^+/+^ [57,58], but only mildly reduced the proliferation rate of PARP1-ΔC ES cells, comparable to PARP1^−/−^ ES cells (Figure 5D). These data suggest that inhibition of PARP1 auto-PARylation that may enrich spontaneous PARP1 trapping at chromatin causes Olaparib-induced cytotoxicity [59]. Both these results demonstrate that PARP1-ΔC trapping does not seem to cause cell death.

### 3.6. Inducible PARP1^ΔC/ind−^ Mice Are Viable but Hypersensitive to Genotoxic Stress

To study the effect of PARP1-iΔC mutation on viability in vivo and evaluate the toxicity of PARP1-ΔC protein on cells, we used PARP1-iΔC mice and MEFs. Both PARP1-iΔC and PARP1-iKO (PARP1 inducible knockout) primary MEFs showed no noticeable difference in cell proliferation, although they proliferated slower compared to PARP1^+/ind−^ counterparts (Appendix A).

To study the response of PARP1-iΔC MEF cells to alkylating agents, a colony formation assay was performed on PARP1^ΔC/ind−^ MEF cells treated with the alkylating agent MNNG, with or without PARPi Olaparib (Appendix A). Interestingly, we did not observe a significant difference in the colony number between PARP1^+/ind−^, PARP1-iΔC, or PARP1-iKO genotypes. However, the sizes of colonies for PARP1^ΔC/ind−^ and PARP1 ^ind−/ind−^ cells were dramatically reduced in response to genotoxic stress (Appendix A), indicating a reduced proliferation, which was further confirmed by a lower proliferation rate for PARP1-iΔC or PARP1-iKO MEF cells compared to controls, after exposure to genotoxic stress (Figure 6A).

To test the genotoxic response of PARP1-ΔC mutation in vivo, we generated inducible PARP1-iΔC mice from the PARP1^ΔC/fl^;CreER^tg/+^ mice after 4 consecutive days of tamoxifen treatment at 8–10 weeks of age. Surprisingly, PARP1-iΔC mice stayed viable and phenotypically normal and survived as long as PARP1^ΔC/+^ or PARP1^+/+^ animals over a period of 6 months. Genotyping confirmed the deletion of the floxed allele (PARP1^ind-^) and the presence of the PARP1-ΔC allele in the liver, lungs, kidneys, brain, spleen, heart, and intestine (Appendix A).

After administering a single intraperitoneal (i.p.) injection of the alkylating agent MNU (150 mg/kg of body weight), we monitored the survival of PARP1-iΔC, PARP1^+/ind−^(wild-type control), and PARP1-iKO mice (positive control, as PARP1^−/−^ mice were hypersensitive to MNU) [13,26]. Compared to the solvent-treated (sham) groups, about 70% of wild-type control mice injected with MNU survived for a month without exhibiting any signs of sickness. Notably, all PARP1-iΔC mice died after six days post-MNU treatment, which was indistinguishable from PARP1-iKO mice (Figure 6B,C). Histological analysis of small intestines revealed a dramatically destroyed intestinal structure of PARP1-iΔC and PARP1-iKO mice by MNU treatment compared to that of the PARP1^+/ind−^ animals (Figure 6D,E). The histological analysis revealed a significant increase in cell death, as judged by TUNEL signals, in both PARP1-iΔC and PARP1-iKO mice compared to PARP1^+/ind−^ controls, which was associated with intestinal atrophy (Figure 6F,G).

## 4. Discussion

The current study reveals that the catalytic activity of PARP1 and 33 very terminal amino acids are essential for mouse development if the scaffold function of PARP1 remains. These crucial features of PARP1 are masked in studies with a complete knockout of PARP1 in mice, which are viable and phenotypically normal during development [26,27]. The lethality of PARP1-ΔC mice might be attributed to the toxic effect of PARP1-ΔC mutation, possibly related to chromatin trapping of PARP1 [9,19] and very likely PARP2 [60] due to the loss of PARylation, protein misfolding and aggregation [61,62], or loss of an essential interactor for the C-terminus [63,64,65].

Remarkably, although the germline PARP1-ΔC mutation causes embryonic lethality, it does not affect the lifespan when the mutation is introduced in adulthood under physiological conditions, thus indicating that the enzymatic activity and the missing C-terminus are critical only at a particular stage of differentiation or for specific organ development. This hypothesis is supported by our observation that although PARP1-ΔC does not compromise ES cell differentiation into primary germ layers, it seems to affect specific organ development, because the mutation caused massive aberrant cystic structures, likely disturbed ECM formation. Cyst formation is a normal developmental process for organ morphogenesis, such as that of the kidneys, liver, pancreas, and others [66]. However, the uncontrolled or unintended formation of cysts is often associated with developmental abnormalities, leading to differentiation defects and pathologies [67,68,69]. It has been shown that mutations in ECM proteins, including laminin and collagen, can contribute to the excessive cyst formation [70,71]. Collagen, for instance, has been demonstrated to play a vital role in cystic structure formation, and mutations of the gene encoding for type IV collagen are associated with increased cyst formation in kidneys [72]. Additionally, deficiency of folliculin, an activator of mTORC1/2 and WNT signalling cascades, alters ECM formation and thereby disrupts alveolar formation followed by cystic lesions [73,74]. PARP1 is a well-known cofactor for many transcription factors [10]. The low amount of PARP1-ΔC and the lack of the C-terminus, which perhaps loses its engagement with critical interactors, may alter expression of genes—for example, in the extracellular matrix (e.g., ECM)—which could be responsible for the cyst phenotype of PARP1-ΔC EBs.

The C-terminal catalytic domain of PARP1 contains the ART domain for NAD^+^ catalysis, and deleting the C-terminus would abolish the PARylation activity [4,5]. Correlating with this expectation, we found a significantly lower level of PAR in PARP1-ΔC cells, compared to wild-type cells, under steady-state conditions, reversely correlating with a high level of intracellular NAD^+^ in these mutant cells. Intriguingly, we detected PARylation activity (judged by PAR formation and NAD^+^ consumption, as well as by PARPi repression) in PARP1-ΔC cells after genotoxic treatment, which nevertheless is higher than that in PARP1^−/−^ cells. This activity is most likely produced by PARP2, which is upregulated in PARP1-ΔC cells, but not PARP1-KO cells. In this regard, it is noteworthy that PARP1-ΔC cells exhibit elevated PARP2 levels even under steady-state conditions, suggesting a compensatory mechanism to cope with the loss of PARylation activity of PARP1-ΔC protein. It is plausible that PARP2 can heterodimerise with PARP1-ΔC protein, probably compensating for PARP1-PARP1 homodimerisation because of low levels of PARP1-ΔC, which is sufficient to perform PARylation [75]. Since we did not detect an apparent increase in PAR levels in response to genotoxic treatment in PARP1 knockout cells (PARP1^−/−^ and PARP1-iKO), it strongly argues for the necessity of PARP1 in PARP2 activation in response to genotoxic stress [75,76].

The PARP1-ΔC protein is expressed at very low levels in PARP1-ΔC cells and, interestingly, PARP1-ΔC protein cannot be ectopically expressed to the wild-type level in PARP1 KO cells. This striking phenotypical feature suggests that low PARP1-ΔC protein levels render cells a survival advantage. The toxicity might be attributed to a “trapping” of PARP1-ΔC protein. However, we did not observe any visible trapping of PARP1-ΔC protein on chromatin in mutant cells under physiological conditions. Strikingly, PARP2, but not PARP1-ΔC, is permanently present at chromatin likely acting as a blocker at chromatin for DNA repair and transcription, as Zha’s laboratory reported [60]. Although PARP1-ΔC association with chromatin is mildly increased after genotoxic treatment, it was not further increased by PARPi Talazoparib, which contrasts with full-length PARP1. Under genotoxic conditions, PAR levels are increased in PARP1-ΔC cells, most likely produced by other ARTs, for example, PARP2, but insufficiently to remove PARP1-ΔC (plausibly in complex with PARP2) from chromatin. Possibly, PARPi inhibits mainly the PARylation activity of PARP1, but does not affect the retention of PARP1-ΔC protein at chromatin, since it lacks the C-terminal catalytic domain. In this regard, PARP inhibitors killed the cells containing wild-type PARP1 (e.g., PARP1^+/+^ and PARP1^+/ΔC^ cells), but surprisingly not PARP1-ΔC cells, demonstrating that PARP1-ΔC cells are protected from the toxic trapping effect of wild-type PARP1 protein [9,59,77]. Altogether, our findings demonstrate that the PARP1-ΔC mutation is detrimental, probably unrelated to its own trapping, but perhaps (though not exclusively) by engaging PARP2 association with chromatin, which is abnormally upregulated in PARP1-ΔC cells. PARP2 is shown to directly bind to DNA by PARPi via the interaction of the WGR domain with DNA and suppressing the PARP1-dependent exchange of PARP2 [60]. It is plausible that PARP1-ΔC by its heterodimerisation with PARP2, but due to lacking PARylation activity, might retain PARP2 on chromatin, which seems cause defects in specific cell types or morphogenesis of embryonic development. However, the direct role of “PARP2 trapping” that blocks other repair factors in DNA repair thereby inducing cell death requires further investigation. Of note, like PARP1 knockout mice, PARP2 complete knockout is compatible with mouse development; however, PARP1–PARP2 double knockout embryos die at gastrulation [78,79]. In this regard, it is worth mentioning that a double knockout, but not a single knockout, of Tankyrase1 or 2, other known PARylation PARPs, also causes mouse embryonic lethality at E10 [80]. These genetic studies highlight a necessity of an effective PARylation for mouse development.

Although cells containing only PARP1-ΔC protein are viable, they show mild defects in DDR, judged by the small colony size in the colony formation assay, indicating a replication defect. Remarkably, PARP1-iΔC mice are viable but highly vulnerable to alkylating agents, behaving like PARP1 complete knockout mice. These phenotypes are reminiscent of a hypo-PARylation mutation PARP1^D993A/D993A^, which impairs the kinetics and branching of PAR formation. However, PARP1^D993A/D993A^ mice have normal development and adult life. Similar to our PARP1-ΔC cellular and mouse models, PARP1^D993A/D993A^ mutant cells exhibit only mild DDR defects, such as replication stress and senescence; PARP1^D993A/D993A^ mice are extremely vulnerable to alkylating agents [13]. Since inhibition of PARP1 activity is a well-known strategy for cancer treatment, our data suggest that high-affinity binding compounds targeting the PARP1 C-terminus, which disrupt the interaction with PARP1 C-terminal interactors, might serve as an alternative or supplement for PARP inhibitors in cancer treatment.

These studies demonstrate that the modulation of the catalytic activity of PARP1, while maintaining PARP1 as a scaffold, can reveal the importance of its PARylation capacity in a range of cellular processes, such as cell death, differentiation, and DDR. Our findings support this, demonstrating that PARP1 protein without enzymatic activity is detrimental to development. While enzymatic activity is not necessary for most organs under normal conditions in adulthood, it remains crucial for acute genotoxic responses. Finally, PARP1-ΔC cellular and mouse models would benefit investigation on how PARP1 lacking the catalytic activity influences chromatin remodelling, transcription activation, and inflammatory responses [10,11,81].

## Figures and Tables

**Figure 1 cells-12-02078-f001:**
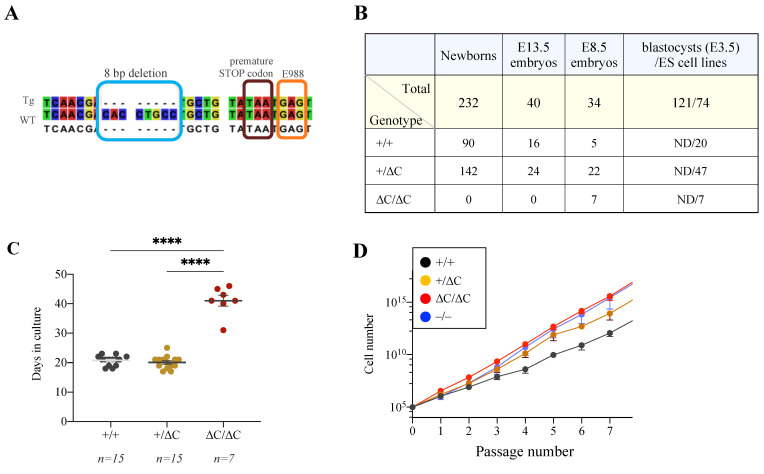
Generation of PARP1^ΔC/ΔC^ mice. (**A**) DNA sequencing of mutant mice revealed an 8 bp deletion that generated a premature stop codon before the codon for E988 amino acid. This resulted in 33 amino acids truncation at the C-terminus of PARP1. The allele is designated as PARP1-ΔC. (**B**) The genotype distribution of offspring from heterozygote breeding. The genotypes of blastocysts (E3.5) were not determined (ND), while the genotypes of the ES cell line derived from them are presented. (**C**) The duration of ES cell line establishment (measured by the number of days in culture from the blastocyst to the formation of a proliferating cell line) for cells with different genotypes is shown. n: the number of ES cell lines of each genotype. (**D**) Accumulative cell number of the ES cell lines of the indicated genotype was counted at the indicated passages in the culture. −/−: PARP1 knockout. Statistics by ordinary one-way ANOVA were used, **** *p* < 0.0001.

**Figure 2 cells-12-02078-f002:**
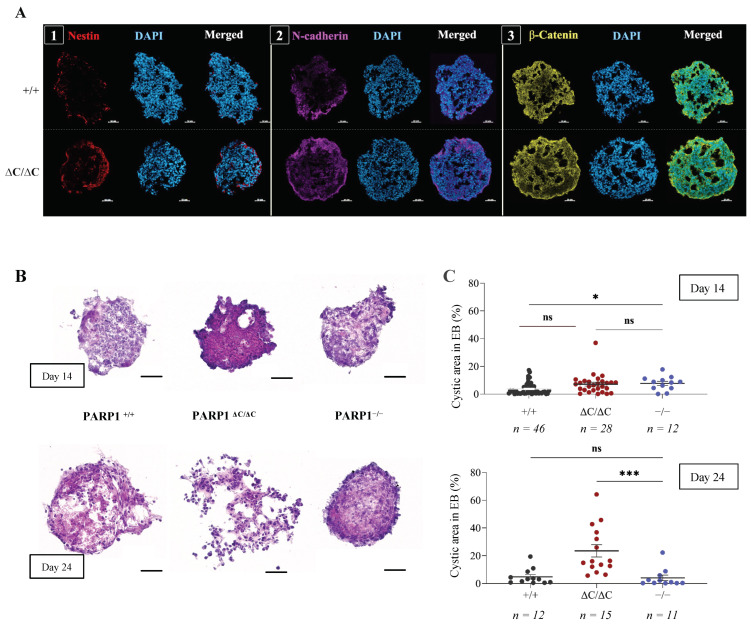
The differentiation assay of ES cells in the embryoid body (EB) formation assay. (**A**) Immunofluorescence staining of embryoid bodies (EBs) from PARP1^+/+^ and PARP1^ΔC/ΔC^ ES cells. EBs at day 14 were sectioned and stained for markers of different germ layers and pluripotency. (1) Nestin: an ectoderm marker; (2) N-cadherin: a mesoderm marker; (3) β-catenin: an endoderm marker. (**B**) The representative images of EBs of indicated genotypes at day 14 and 24 of EB formation. The cryosections were stained with HE. Scale bar: 50 μm. *n* = 3. (**C**) The quantification (percentage) of the cyst area in EBs of day 14 and 24. *n* = 3. Statistics by ordinary one-way ANOVA were used, * *p* < 0.05, *** *p* < 0.001, ns, not significant.

**Figure 3 cells-12-02078-f003:**
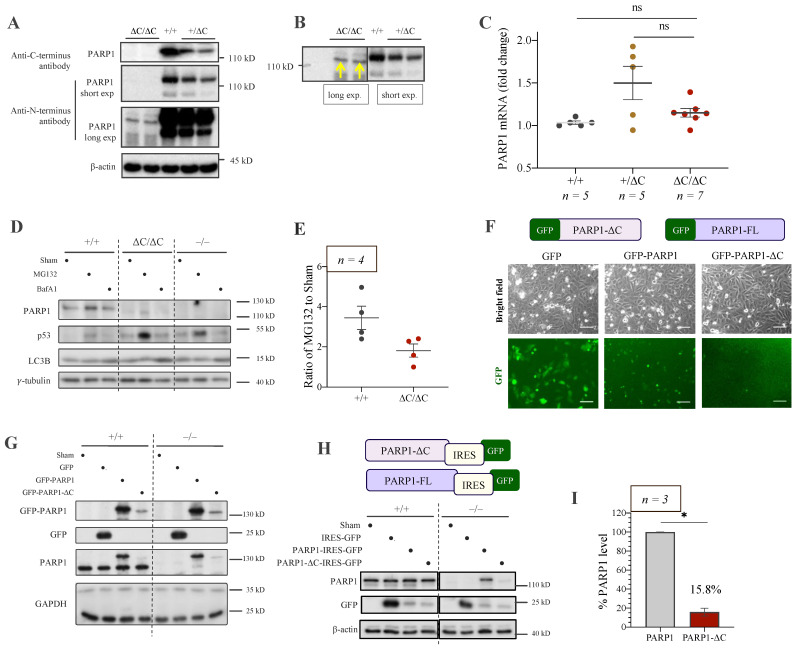
Analysis of PARP1-ΔC protein expression. (**A**) Western blot analysis of ES cells of the indicated genotypes by PARP1 antibody (C2-10) against the C-terminal PARP1 revealed no PARP1 protein in PARP1^ΔC/ΔC^ ES cells. Polyclonal antibody against PARP1 N-terminus (DEVD) detected a shorter PARP1-ΔC at low levels. β-actin is a loading control. (**B**) The overlay of an over-exposed film with a short-exposed film from (**A**), blotted by the antibody against the DEVD sequence. Yellow arrows indicate the shorter PARP1-ΔC protein band. (**C**) RT-qPCR analysis of PARP1 mRNA levels in ES cells of the indicated genotypes. *n* = 5. (**D**) Western blot analysis of PARP1-ΔC protein levels in ES cells compared to PARP1^+/+^ and PARP1^−/−^ after treatment with proteasome inhibitor MG132 (10 μM, 4 h) or lysosome inhibitor BafilomycinA1 (BafA1, 0.8 μM, 4 h). p53 and LC3B are used for the control of protein stability through the proteasome and lysosome pathway, respectively. *γ*-tubulin is a loading control. (**E**) Quantification of PARP1-FL and PARP1-ΔC protein levels in ES cells after treatment with MG132 or BafA1. The protein levels after MG132 treatments were normalised to sham controls. Statistics by ordinary one-way ANOVA were used; ns, not significant. (**F**) The scheme of the GFP-PARP1 (full length) and GFP-PARP1-ΔC constructs used for the experiment. Representative images of PARP1^−/−^ U2OS cells 24 h after transfection with the indicated vectors in a bright field (upper panel) and GFP channel (lower panel). Scale bar: 100 μm. (**G**) Western blot analysis of ectopic PARP1-ΔC expression compared to PARP1-FL and GFP in PARP1^+/+^ or PARP1^−/−^ U2OS cells. GAPDH serves a loading control. *n* = 4. (**H**) The scheme of two PARP1 expression constructs. The IRES element was used for independent control of the transfection efficiency. Western blot analysis of ectopic PARP1-ΔC expression compared to PARP1-FL in PARP1^+/+^ or PARP1^−/−^ HeLa cells. GFP expression is an independent control for transfection. β-actin is a loading control. *n* = 3. (**I**) Quantification of PARP1-FL and PARP1-ΔC protein levels in PARP1^−/−^ HeLa after normalisation to GFP from (**H**). *n* = 3. Statistics by Mann–Whitney test were used, * *p* < 0.05.

**Figure 4 cells-12-02078-f004:**
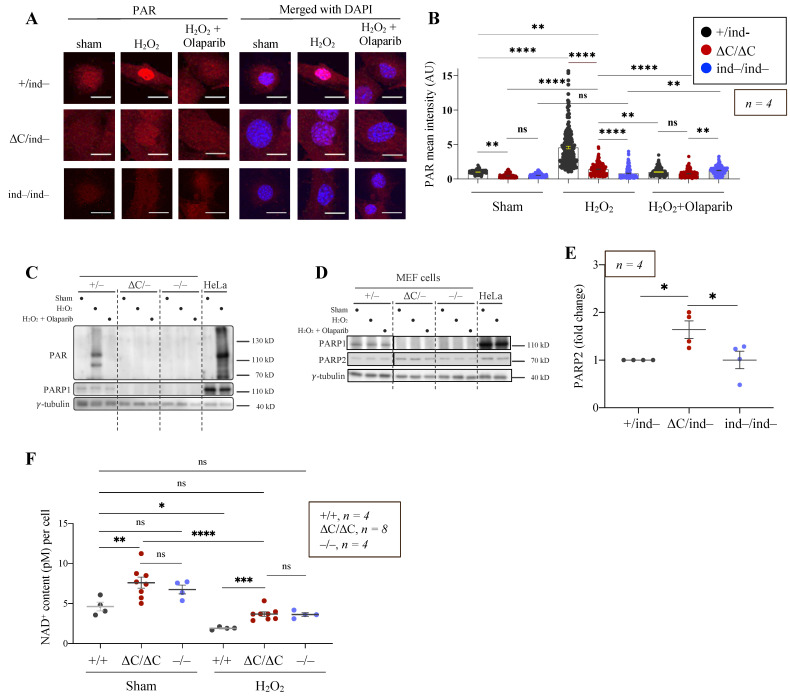
The catalytic activity of PARP1-ΔC cells. (**A**) Immunofluorescent staining of PAR in MEFs of the indicated genotypes without (sham) or with the H_2_O_2_ treatment (1 mM, 15 min) in the presence or absence of PARP inhibitor Olaparib (10 μM, 2 h). Scale bar: 20 μm. *n* = *8*. (**B**) Quantification of PAR intensity, *n* = 4. Statistics by ordinary one-way ANOVA were used; ** *p* < 0.01, **** *p* < 0.0001, ns, not significant. (**C**,**D**) Western blot analysis of MEF cells of the indicated genotypes without (sham) or with the H_2_O_2_ treatment (1 mM, 15 min) in the presence or absence of Olaparib (10 μM, 15 min). The antibodies used are indicated. γ-tubulin is a loading control. *n* = 4. (**E**) Quantification of PARP2 protein levels in MEFs, normalised to γ-tubulin. The fold change is normalised to PARP2 levels in PARP1^+/ind−^ genotype. *n* = 4. Statistics by ordinary one-way ANOVA were used; * *p* < 0.05. (**F**) Quantification of NAD^+^ levels ES cells of the indicated genotype without (sham) or with the H_2_O_2_ treatment of (1 mM, 15 min). *n* = 4. Statistics by ordinary one-way ANOVA were used; * *p* < 0.05, ** *p* < 0.01, *** *p* < 0.001, **** *p* < 0.0001, ns, not significant.

**Figure 5 cells-12-02078-f005:**
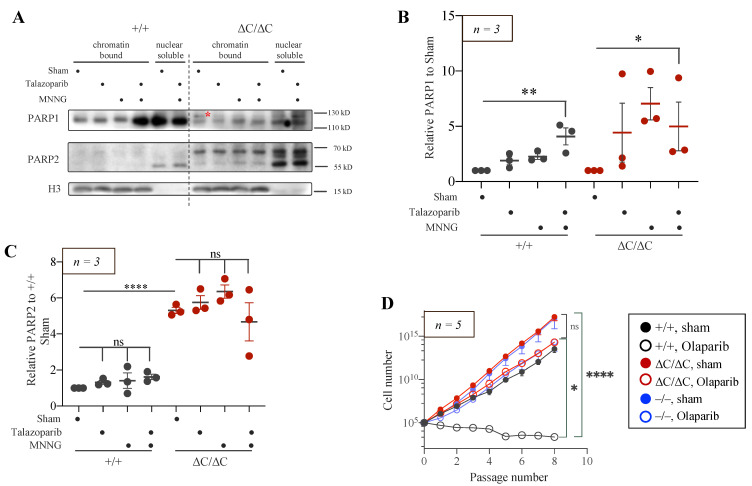
Trapping of PARP1-ΔC and PARP2 proteins under genotoxic stress in ES cells. (**A**) Western blot analysis of a chromatin-bound fraction of PARP1-FL, PARP1-ΔC, and PARP2 proteins with MNNG (100 μM, 15 min) and/or PARP inhibitor Talazoparib (10 μM, 15 min). H3 is a loading control. *n* = 3. The red star indicates a non-specific band. (**B**,**C**) Quantification of PARP1-FL, PARP1-ΔC, and PARP2 protein levels after MNNG treatment and/or Talazoparib. The protein levels were normalised to H3 and then to sham controls. Statistics by ordinary one-way ANOVA were used; * *p* < 0.05, ** *p* < 0.01, **** *p* < 0.0001, ns, not significant. (**D**) The proliferation of ES cell lines in the presence or absence of Olaparib (PARPi, 2 μM). *n* = 5. Statistics by one-way repeated measures nonparametric test, matched rows with Friedman test were used to analyse the mean values within individual days, * *p* < 0.05, **** *p* < 0.0001.

**Figure 6 cells-12-02078-f006:**
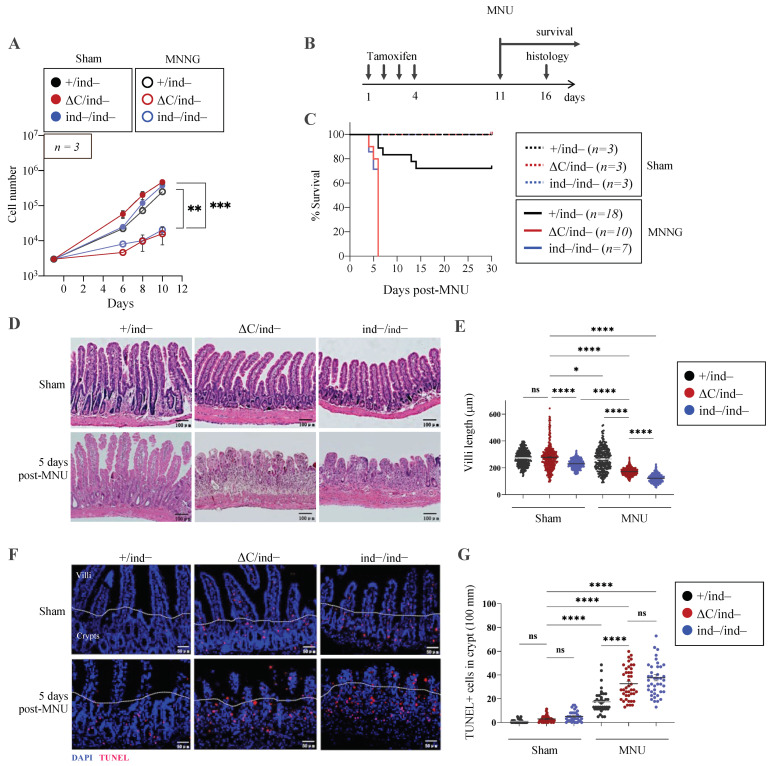
Inducible PARP1^ΔC/−^ cells and mice are viable but sensitive to DNA damage. (**A**) Accumulative proliferation curves of immortalised MEF cells. Three thousand cells of the indicated genotype were plated in the 6-well plate in 3 replicates. The next day, cells were treated with sham or with MNNG (8 μM, 1 h) (day 0). On day 1, cells were re-passaged, and the number of cells was determined on days 6, 8, and 10 in culture, *n* = 3. (**B**) The scheme for mice (6–8 weeks old) treated with tamoxifen (75 mg/kg/day, for 4 days) by intraperitoneal injections. Seven days later (day 11), the mice were injected with a single dose of MNU (150 mg/kg) or solvent (sham), and survival was evaluated. Five days after MNU treatment (day 16), histology of the intestine was performed. (**C**) The Kaplan–Meier survival curves of mice with the indicated genotype treated with solvent (sham) or MNU. The number (*n*) of mice in each group is indicated. Note, +/ind- genotype is a wild-type control does not appear clearly in the graph. (**D**) Representative images of small intestine sections 5 days after MNU treatment. The cryosections were stained with HE. Scale bar: 100 μm. (**E**) Quantification of villi length in the small intestine (3 mice per group; each dot represents the length of one villus). (**F**) Representative images of small intestine cryosections of the indicated genotype 5 days after MNU treatment stained against TUNEL. Scale bar: 50 μm. (**G**) Quantification of TUNEL signal of the small intestine. (3 mice per group, each dot represents a number of TUNEL-positive cells per 100 μm intestine scored). Statistics by ordinary one-way ANOVA were used; * *p* < 0.05, ** *p* < 0.01, *** *p* < 0.001, **** *p* < 0.0001, ns, not significant.

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
