# Peer review of "Poly(ADP-Ribose) Polymerase-1 Lacking Enzymatic Activity Is Not Compatible with Mouse Development"

_cells, 2023, doi:10.3390/cells12162078_

Round 1

Reviewer 1 Report

In their current work, Kamaletdinova et al. investigated the impact of a PARP1 hypomorph mutation (truncated catalytically inactive PARP form that express on a significantly lower level) on mouse development and the cells' sensitivity to genotoxic stress. The findings of this study hold significant importance and make a valuable contribution to the field. They provide evidence for the crucial role of PARP1 in the developmental process. However, the conclusions drawn from this study can be misleading. The main outcome is that while the PARP1 mutation causes early developmental arrest, it is not necessary for adult animals. The logical implication of this study is that the protein is required specifically for development. The substantial amount of data published previously demonstrates that PARP1 is essential for proper gene expression during development. Studies conducted on Drosophila, where PARP is the sole nuclear poly(ADP-ribosyl)ation enzyme, have highlighted its critical role in development. In mice, double PARP1 and PARP2 knockouts result in early developmental lethality, with arrest during gastrulation. All these indicates that poly(ADP-ribosyl)ation is crucial for developmental process. For the other hand, many proteins, that are required for DNA repair don’t cause any problems during development.

The possible explanation for the observed phenotypes in the current study is as follows: in regular PARP1 knockouts, PARP2 assumes its role in chromatin, occupying the promoters and participating in the activation and expression of developmental genes. However, in the studied mutant, the truncated PARP1 occupies the necessary positions in chromatin, but since it is catalytically inactive, it fails to activate the genes.

1.      I have concerns regarding the selection of markers for different cell lineages. Specifically, the presence of N-cadherin in mesoderm at day 14 of EB formation seems incorrect. Could you provide the references that guided the choice of current markers? By day 14 of differentiation, various tissue types should already be present, and N-cadherin should be found in neuroepithelial cells instead. Additionally, I have doubts about using beta-catenin as an endoderm marker, especially at day 14 of differentiation. The staining quality appears blurry and unusual. It seems that beta-catenin stains all cells (is it a non-specific staining?), while nestin is hardly visible.

On the other hand, the morphological differences between mutant and control EBs are quite evident in regular H&E staining. Furthermore, according to Figure S2 (if I correctly interpret the two asterisks), the level of N-cadherin shows a significant difference, although this fact is not mentioned in the text. Moreover, the text states that mRNA levels were similar. Consequently, I am not convinced that the differentiation studies were sufficient and conducted accurately.

Considering that hemizygously mutated embryos are present on day E8.5, it would be highly beneficial to examine their morphology and potential developmental abnormalities.

2.      My primary concerns revolve around the concept of PARP2 "trapping" to chromatin. Its confirmation is lacking. Mere fractionation and the presence of a protein in the "chromatin" fraction do not prove that it will have any impact on cells or development. Further direct experiments are necessary to demonstrate that PARP2 "trapping" indeed has an effect. It is completely unclear how this so-called PARP2 trapping can affect development but not have any impact on other differentiated cells in an adult organism. If it does somehow affect cell proliferation, problems would be expected in adult organisms where cell proliferation also occurs. Therefore, I suggest completely omitting this part of the paper unless more direct evidence is provided.

Minor:

1.      The quality of images should be improved, too pixelized and blurry.

2.      Undetectable scale on histology slides

3.      PARP1-/- cells, especially ES cells, still have pADPr (line 593). This amount is lower than in WT cells but still present.

4.      It needs to be mentioned for how many generations they outcrossed the PARP1dC/+ mouse line. It is possible to have the off-target mutation with gRNA that has such a low off-target score. Are there any predicted off-targets close to PARP1 on the same chromosome that could be transmitted in one crossingover group with PARP1?

5.      Check if it’s a mistake in gRNA: aaggatactcattatacagcaggCagg

6.      Some antibodies that were used are not present in M&M section. Particular interest in PARP1 C- and N-terminal domain antibodies. Also: p53, GFP, LC3B.

7.      Fig5 fix PRP2 to PARP2

Author Response

Reviewer#1

In their current work, Kamaletdinova et al. investigated the impact of a PARP1 hypomorph mutation (truncated catalytically inactive PARP form that express on a significantly lower level) on mouse development and the cells' sensitivity to genotoxic stress. The findings of this study hold significant importance and make a valuable contribution to the field. They provide evidence for the crucial role of PARP1 in the developmental process. However, the conclusions drawn from this study can be misleading. The main outcome is that while the PARP1 mutation causes early developmental arrest, it is not necessary for adult animals. The logical implication of this study is that the protein is required specifically for development. The substantial amount of data published previously demonstrates that PARP1 is essential for proper gene expression during development. Studies conducted on Drosophila, where PARP is the sole nuclear poly(ADP-ribosyl)ation enzyme, have highlighted its critical role in development. In mice, double PARP1 and PARP2 knockouts result in early developmental lethality, with arrest during gastrulation. All these indicates that poly(ADP-ribosyl)ation is crucial for developmental process. For the other hand, many proteins, that are required for DNA repair don’t cause any problems during development.

The possible explanation for the observed phenotypes in the current study is as follows: in regular PARP1 knockouts, PARP2 assumes its role in chromatin, occupying the promoters and participating in the activation and expression of developmental genes. However, in the studied mutant, the truncated PARP1 occupies the necessary positions in chromatin, but since it is catalytically inactive, it fails to activate the genes.

We appreciate this reviewer’s positive view and comments. His/her critical comments are helpful to improve the quality of the manuscript.

  1. I have concerns regarding the selection of markers for different cell lineages. Specifically, the presence of N-cadherin in mesoderm at day 14 of EB formation seems incorrect. Could you provide the references that guided the choice of current markers? By day 14 of differentiation, various tissue types should already be present, and N-cadherin should be found in neuroepithelial cells instead. Additionally, I have doubts about using beta-catenin as an endoderm marker, especially at day 14 of differentiation. The staining quality appears blurry and unusual. It seems that beta-catenin stains all cells (is it a non-specific staining?), while nestin is hardly visible.

It should be mentioned that the EB formation may not simply refer to mouse development in vivo. Day 14 of EB formation under non-stress or without inducers of differentiation allows us to test the general differentiation capacity of PARP1-DC ES cells into primary germ-layers and cannot simply study differentiation of specific cell lineages, if no specific inducers used. Therefore, the choice of the markers for IF staining is purely to test primary germlayer derived from these ES cells. N-cadherin is a well-known marker for mesoderm (Li et al. EMBO J 2015; Warga and Kane, Dev Biol 2007). β-catenin, as a part of canonical WNT signalling in the absence of LIF, is known to activate primitive endoderm genes Price et al. Stem Cell 2013. Moreover, β-catenin is a key molecule in gastrulation, responsible for definitive endoderm formation (Engert et al. Development 2013, Mukherjee et al. eLife 2020). Thus, β-catenin was chosen as an endoderm marker. We added references in the revised text for the choice of these markers (page 8, lines 341-342). Nevertheless, we acknowledge that these markers would not allow us to pinpoint down the specific cell types which are affected by PARP1-DC mutation.

We apologize for the suboptimal quality of images, but it might stem from the conversion of original figures to merged files during submission.

On the other hand, the morphological differences between mutant and control EBs are quite evident in regular H&E staining. Furthermore, according to Figure S2 (if I correctly interpret the two asterisks), the level of N-cadherin shows a significant difference, although this fact is not mentioned in the text. Moreover, the text states that mRNA levels were similar. Consequently, I am not convinced that the differentiation studies were sufficient and conducted accurately.

We apologize for our error and inaccuracy. Indeed Fig S2 showed different expression of N-cadherin in PARP1-/- EBs compared to PARP1-DC and control counterparts. We meant that there was no difference between PARP1-DC and control cells. Now we have corrected the description (page 8, lines 345-347).

Considering that hemizygously mutated embryos are present on day E8.5, it would be highly beneficial to examine their morphology and potential developmental abnormalities.

Thanks for this suggestion. We agree that a detailed developmental study would allow us to learn what developmental defects of PARP1-DC embryos. Scientifically, it is interesting to further study development in PARP1-DC embryos and at adult stage of inducible PARP1-DC animals, which require a huge amount of mouse work and cannot performed at this stage. Importantly, we hope that the reviewer would agree that these experiments represent a separate study, which cannot be accommodated in the current study.

  1. My primary concerns revolve around the concept of PARP2 "trapping" to chromatin. Its confirmation is lacking. Mere fractionation and the presence of a protein in the "chromatin" fraction do not prove that it will have any impact on cells or development. Further direct experiments are necessary to demonstrate that PARP2 "trapping" indeed has an effect. It is completely unclear how this so-called PARP2 trapping can affect development but not have any impact on other differentiated cells in an adult organism. If it does somehow affect cell proliferation, problems would be expected in adult organisms where cell proliferation also occurs. Therefore, I suggest completely omitting this part of the paper unless more direct evidence is provided.

Thank you for the comment and discussion. I agree that we found that substantial binding of PARP2 in chromatin in PARP1-DC cells in contrast to wildtype controls. We acknowledge that we do not have direct evidence showing that the enrichment of PARP2 at chromatin is responsible for development defects of these mutant embryos. As the reviewer correctly pointed out that this may not reconcile the facts that adult tissues cope well with PARP1-DC mutation. The finding that PARP2 trapping occurred in these PARP1-DC mutant cells is interesting, which we feel should be documented. However, we tried not to overstate our observations and have carefully checked our presentation (page 18, lines 629-630). We also added extra discussion on possible involvement of PARP2, in conjunction with the comment of Reviewer #2-point 2 (page 17, lines 601-604; page 18, lines 622-629).

Minor:

  1. The quality of images should be improved, too pixelized and blurry.

The quality of images in the merged manuscript might have been stemmed from the uploaded figs. We will ensure to have high quality images in the final publication.

  1. Undetectable scale on histology slides

Thank you for pointing out. These are all fixed.

  1. PARP1-/- cells, especially ES cells, still have pADPr (line 593). This amount is lower than in WT cells but still present.

Yes, a trace amount of pADPr in PARP1-/- cells might be expected, likely due to PARP2, which has been known well in the field (Shieh, et al. JBC 1998; see also reviews Luo & Krauss. G&D 2012; Kamaletdinova et al. Cells 2019).

  1. It needs to be mentioned for how many generations they outcrossed the PARP1dC/+ mouse line. It is possible to have the off-target mutation with gRNA that has such a low off-target score. Are there any predicted off-targets close to PARP1 on the same chromosome that could be transmitted in one crossingover group with PARP1?

The founder was already produced in C57BL/6JRj congenic line. Both in year 2015 and now very recently, the off-target were estimated with a score ≥1 according to Doench et al. (Nat Biotech 2016). We crossed 5 generations of PARP1-DC with wildtype and also many generations with heterozygous mutants. In addition, these mutant mice were also crossed with PARP1-/- and PARG110 ko mice (data not shown). Overall, the lethal phenotype has not been rescued in any individual animals who carry the PARP1-DC allele regardless of genetic background or modifications. These genetic studies altogether demonstrate that PARP1-DC mutation caused embryonic lethality and the off-target effect is likely not the case.

  1. Check if it’s a mistake in gRNA: aaggatactcattatacagcaggCagg

We apologize for the typo about the sgRNA. It is corrected as

AAGGATACTCATTATACAGCAGG (page 3, lines 87-88).

  1. Some antibodies that were used are not present in M&M section. Particular interest in PARP1 C- and N-terminal domain antibodies. Also: p53, GFP, LC3B.

We apologize for missing information and now they are included (page 6, lines 277-281).

  1. Fig5 fix PRP2 to PARP2

Fixed.

Reviewer 2 Report

PARP1 plays a central role in DNA damage signaling and transcriptional regulations only to name two of its many cellular roles. In 1995 Wang et al. (senior author of this MS) generated the first PARP1 knockout mice which was a very important milestone in PARylation research and the whole field benefited greatly from the use of these mice and cells derived from them. This time the senior author teamed up with another leader of the field (M. Hottiger) to generate a mouse strain with catalytically inactive PARP1 in order to distinguish between the roles of PARP1 requiring enzyme activity from the roles that depend on the scaffolding function of the protein. The plan was to introduce a E988->K988 mutation, which, based on previous biochemical studies, was expected to yield a catalytically inactive PARP1. However, genome editing resulted in an 8 bp deletion causing a frameshift and, due to a premature STOP codon, a 33 AA truncation of the protein. This PARP1-deltaC protein indeed lacked enzyme activity and surprisingly resulted in an embryonic lethal phenotype. The authors characterized this phenotype in detail and suggest that an upregulated and chromatin trapped PARP2 might contribute to lethality. Moreover, the authors also generated an inducible (adult) version of PARP1-deltaC with a viable phenotype and investigated the genotoxic stress response in their cells.

Although the discovery process took some unexpected turns, nevertheless the findings are very interesting and thought provoking for scientists working in the field. The amount and quality of the data generated in this study clearly warrant publication of these findings.

Minor points

1. It was not clear why authors used olaparib PARP inhibitor in some experiments and talazoparib in others. Please justify.

2. There aren't too many PARP family members that function as bona fide PARPs (i.e. are capable of synthesizing PAR polymer). Therefore, in the Discussion, authors might want to consider covering their potential roles in this embryonic lethal phenotype even if they were not investigated in this study. 

3. Considering that PARPi are used for cancer treatment, authors might want to give an insight into what practical consequences of their results they consider possible.

The paper is concide and well written. I only found few sentences that may require rephrasing. (E.g. line 598 "The toxicity was thought by a trapping...")

Author Response

Reviewer#2

PARP1 plays a central role in DNA damage signaling and transcriptional regulations only to name two of its many cellular roles. In 1995 Wang et al. (senior author of this MS) generated the first PARP1 knockout mice which was a very important milestone in PARylation research and the whole field benefited greatly from the use of these mice and cells derived from them. This time the senior author teamed up with another leader of the field (M. Hottiger) to generate a mouse strain with catalytically inactive PARP1 in order to distinguish between the roles of PARP1 requiring enzyme activity from the roles that depend on the scaffolding function of the protein. The plan was to introduce a E988->K988 mutation, which, based on previous biochemical studies, was expected to yield a catalytically inactive PARP1. However, genome editing resulted in an 8 bp deletion causing a frameshift and, due to a premature STOP codon, a 33 AA truncation of the protein. This PARP1-deltaC protein indeed lacked enzyme activity and surprisingly resulted in an embryonic lethal phenotype. The authors characterized this phenotype in detail and suggest that an upregulated and chromatin trapped PARP2 might contribute to lethality. Moreover, the authors also generated an inducible (adult) version of PARP1-deltaC with a viable phenotype and investigated the genotoxic stress response in their cells.

Although the discovery process took some unexpected turns, nevertheless the findings are very interesting and thought provoking for scientists working in the field. The amount and quality of the data generated in this study clearly warrant publication of these findings.

Thank you for appreciation of the usefulness of our study.

Minor points

  1. It was not clear why authors used olaparib PARP inhibitor in some experiments and talazoparib in others. Please justify.

Talazoparib induces 100 times stronger trapping than Olaparib (Rudolph et al. PNAS 2022). We first tested Olaparib in our initial survival experiments in order to sufficiently decrease the viability of PARP1-DC ES cells but to preserve for wildtype ES cells. Olaparib is also known to have a toxic effect on some types of stem cells such as HSCs (Herath et al. Front Oncol 2019). Indeed, Talazoparib showed a stronger effect on PARP1 trapping in both PARP1-DC and wildtype genotypes. During the course of experiments, we used both Olaparib and Talazoparib and found similar effects of these inhibitors toward to PARP1-DC mutation.

  1. There aren't too many PARP family members that function as bona fide PARPs (i.e. are capable of synthesizing PAR polymer). Therefore, in the Discussion, authors might want to consider covering their potential roles in this embryonic lethal phenotype even if they were not investigated in this study. 

Thank you for the suggestion. Indeed, among 17-18 (depending on counting 13.1 and 13.2 as one or two family members) PARPs, only PARP1, PARP2, PARP5a and PARP5b (or Tankyrase1 or 2) have been shown to harbor PARylation activity. The rest of the members have MARylation activity or are inactive.

Tankyrase1 or 2 single gene KO in mice are viable, whereas a double KO of Tankyrase1 or 2 caused embryonic lethality Chiang et al. PLOS One 2008), which are similar to the situation of PARPs 1-2 double KO (de Murcia et al. EMBO J 2003). Following the suggestion, we have added additional discussion, in conjunction with the comment of Reviewer #1-point 1 (page 18, lines 622 - 635).

  1. Considering that PARPi are used for cancer treatment, authors might want to give an insight into what practical consequences of their results they consider possible.

We have added some discussion of this point (page 18 lines 644 - 648).

Comments on the Quality of English Language

The paper is concide and well written. I only found few sentences that may require rephrasing. (E.g. line 598 "The toxicity was thought by a trapping...")

We checked and corrected those sentences.

Reviewer 3 Report

In this manuscript, the authors illustrate how the inactivated PARP1 (PARP1ΔC/ΔC) mutation results in embryonic lethality. They additionally determine that PARP1ΔC/ΔC mutant Embryonic Stem (ES) cells anomalously evolve into a high volume of cysts, which insinuates imperfections in the epithelial cells. The evidence portrays an exceedingly low degree of expression of the PARP1-ΔC protein, whilst an augmented and sustained chromatin presence of PARP2 is perceived in PARP1-ΔC cells. This suggests that PARP2 is engaged at the chromatin level by the PARP1-ΔC protein. The researchers further established that the introduction of the PARP1-ΔC mutation in adult mice did not undermine their viability, but it did heighten their susceptibility to alkylating agents. They ultimately determine that the catalytically inactive mutant PARP1 induces a developmental block by facilitating the entrapment of PARP2.

Through a comprehensive investigation, they extrapolate the nonenzymatic role of the PARP1 protein in development, illustrating how such a role can be deleterious without enzymatic activity. The manuscript is cogently written and engaging. I have only trivial recommendations for the authors to consider, and I most emphatically endorse the manuscript's publication upon addressing these minor suggestions.

Specific points to be considered:

1.     The authors notice a substantial augmentation of cyst-like structures among PARP1ΔC/ΔC EBs, commonly associated with developmental abnormalities. It would be worthwhile for them to elaborate on why the PARP1ΔC/ΔC mutation provokes such a cyst manifestation.

2. Please explain the rationale for conducting experiments with different PARPi, specifically Olaparib (Fig 4) and Talazoparib (Fig 5).

33. The labelling of Panel 'B' and 'C' needs to be included in Figure S2.

4. The '"Figure. S2A-D" reference on line 343 should be corrected to read as "Figure. S2A-C".

5. The incorporation of a Molecular Weight marker for the anti-C’ PARP1 blot in Figure 3A would be appreciated.

Author Response

Reviewer#3

In this manuscript, the authors illustrate how the inactivated PARP1 (PARP1ΔC/ΔC) mutation results in embryonic lethality. They additionally determine that PARP1ΔC/ΔC mutant Embryonic Stem (ES) cells anomalously evolve into a high volume of cysts, which insinuates imperfections in the epithelial cells. The evidence portrays an exceedingly low degree of expression of the PARP1-ΔC protein, whilst an augmented and sustained chromatin presence of PARP2 is perceived in PARP1-ΔC cells. This suggests that PARP2 is engaged at the chromatin level by the PARP1-ΔC protein. The researchers further established that the introduction of the PARP1-ΔC mutation in adult mice did not undermine their viability, but it did heighten their susceptibility to alkylating agents. They ultimately determine that the catalytically inactive mutant PARP1 induces a developmental block by facilitating the entrapment of PARP2.

Through a comprehensive investigation, they extrapolate the nonenzymatic role of the PARP1 protein in development, illustrating how such a role can be deleterious without enzymatic activity. The manuscript is cogently written and engaging. I have only trivial recommendations for the authors to consider, and I most emphatically endorse the manuscript's publication upon addressing these minor suggestions.

We thank this reviewer’s positive view of the manuscript and suggestions.

Specific points to be considered:

  1. The authors notice a substantial augmentation of cyst-like structures among PARP1ΔC/ΔC EBs, commonly associated with developmental abnormalities. It would be worthwhile for them to elaborate on why the PARP1ΔC/ΔC mutation provokes such a cyst manifestation.

PARP1 is a well-known cofactor for many transcription factors. The low amount of PARP1-DC and / or missing the C-terminus may alters gene expression. One may blame disturbed genes, for example in extracellular matrix (eg., ECM), which may account for the cyst phenotype of mutant EBs. It is also possible that the major effect on cyst formation could be due to the missing C-terminus and sequentially several critical interactors. PARP1 was shown to bind directly to several types of lipids belonging to the PPIns family (Mazloumi Gavgani et al. Mol Cell Proteo 2021). If missing, the composition of the lipids may lead to improper anchoring of membrane protein and loss of cell-cell interaction. At this stage, we are reluctant to make too much speculative discussion. Nevertheless, we have discussed some of our ideas in the revised text (page 17, lines 584-587).

  1. Please explain the rationale for conducting experiments with different PARPi, specifically Olaparib (Fig 4) and Talazoparib (Fig 5).

This point was also raised by Reviewer #2. Talazoparib induces 100 times stronger trapping than Olaparib (Rudolph et al. PNAS 2022). We first tested Olaparib in our initial survival experiments in order to sufficiently decrease the viability of PARP1-DC ES cells but to preserve for wildtype ES cells. Olaparib is also known to have a toxic effect on some types of stem cells such as HSCs (Herath et al. Front Oncol 2019). Indeed, Talazoparib showed a stronger effect on PARP1 trapping in both PARP1-DC and wildtype genotypes. During the course of experiments, we used both Olaparib and Talazoparib and found similar effects of these inhibitors toward to PARP1-DC mutation.

  1. The labelling of Panel 'B' and 'C' needs to be included in Figure S2.

Original A-C are merged as single citation. It is corrected in the revised text.

  1. The '"Figure. S2A-D" reference on line 343 should be corrected to read as "Figure. S2A-C".

Thank you for pointing out this. It is corrected as “Fig S2”.

  1. The incorporation of a Molecular Weight marker for the anti-C’ PARP1 blot in Figure 3A would be appreciated.

Based on the blot, we cannot mark clearly the molecular weight of PARP1-DC, but it should be about 2.7kD less than wildtype PARP1.

Round 2

Reviewer 1 Report

The current work's conclusions and model lack support from experimental data. While adult mice with PARP mutation show normal viability, embryonic development is arrested. It is unclear how PARP mutation affects cell viability when all cells in the adult mice appear normal. The term "PARP2 trapping" is misleading; experiments merely demonstrate higher presence of PARP2 in the chromatin fraction under all conditions, but it doesn't imply actual "trapping." The cited paper (60) also lacks direct evidence of such "trapping" and relies solely on immunofluorescence staining experiments. The authors must either provide direct experimental data on "trapping" to DNA or remove this misleading section.

The statement "These genetic studies highlight a sufficient and dynamic PARylation for cell survival" is incorrect. These studies actually suggest that effective PARylation is necessary for development, but not specifically for cell survival.

Author Response

Reply:

It is not our intention to conclude that the enrichment of PARP2 at chromatin is responsible for developmental phenotypes, but rather to give readers our thoughts in discussion. As the matter of the fact, the term ”trapping” is widely used to explain (one of explanations) PARP inhibitor-killing effect of cancer cells (which are defective in one of DNA repair pathways). We observed a spontaneously high level of PARP2 at chromatin, which could not be further increased by genotoxic treatment and PARP inhibitors (which otherwise induces wildtype PARP1 “trapping”). These facts allowed us to think that PARP2 is “associated” with chromatin, engaged by PARP1-delC, behaving as “trapping”, a phenomenon described for PARP1 when it cannot be released from chromatin by auto-PARylation. Therefore, “trapping” is mainly used as semantic expression and double quoted in our Discussion. In our “Results” session, we have tried to carefully phrase our presentation and used “enrichment”, “retention” or “association” of PARP2 at the chromatin.

We found that the enrichment of PARP2 is interesting and discussed possible explanations using Ref 60, which documented interesting findings (immunofluorescent staining is a routine and well accepted method for locating proteins in subcellular compartments). Although it is technically challenging to test a direct role of “PARP2 trapping” in cell death in our experimental setting (PARP1-delC cells), we would like to document these findings in the current manuscript, because they should be known by the field. Nevertheless, we weakened our claim by deleting a sentence (lines 492-494) and acknowledged a need for studying “PARP2 trapping” in future studies (line 628-629).

The statement "These genetic studies highlight a sufficient and dynamic PARylation for cell survival" is incorrect. These studies actually suggest that effective PARylation is necessary for development, but not specifically for cell survival.

Reply:

Indeed, “cell survival” can be a reason for developmental arrest—any reasons / cellular mechanisms eventually would have to go through cell death to arrest development, alter organ formation, and eventually cause embryo lethality. To be precise, we changed the sentence as “These genetic studies suggest a necessity of an effective PARylation for development" (lines 633-634).
